# Antibacterial Efficacy of Cold-Sprayed Copper Coatings against Gram-Positive *Staphylococcus aureus* and Gram-Negative *Escherichia coli*

**DOI:** 10.3390/ma14226744

**Published:** 2021-11-09

**Authors:** Novana Hutasoit, Sanjida Halim Topa, Muhammad Awais Javed, Rizwan Abdul Rahman Rashid, Enzo Palombo, Suresh Palanisamy

**Affiliations:** 1School of Engineering, Swinburne University of Technology, Hawthorn 3122, Australia; nhutasoit@swin.edu.au (N.H.); spalanisamy@swin.edu.au (S.P.); 2DMTC Ltd., Hawthorn 3122, Australia; 3School of Science, Computing and Engineering Technologies, Swinburne University of Technology, Hawthorn 3122, Australia; stopa@swin.edu.au (S.H.T.); mjaved@swin.edu.au (M.A.J.); epalombo@swin.edu.au (E.P.)

**Keywords:** cold spray, copper, coating, antibacterial property, *E. coli*, *S. aureus*, galvanic potential

## Abstract

Contact surfaces have been identified as one of the main routes for pathogen transmission. The efficacy to kill both viruses and bacteria on touch surfaces is critical to reducing the rampant spread of harmful pathogens. Copper is one such material that has been traditionally used for its antimicrobial properties. However, most contact/touch surfaces are made up of steel or aluminum due to their structural properties. Therefore, coating high-touch components with copper is one possible solution to improve antibacterial efficacy. In this study, copper was coated on both stainless steel and aluminum substrates using a cold spray process which is a fast and economic coating technique. The coated samples in both as-deposited and heat-treated states were exposed to *Escherichia coli* and *Staphylococcus aureus* bacteria, and their efficacy was compared with bulk copper plate. It was found that both bacterial cells responded differently to the different coating properties such as coating thickness, porosity, hardness, surface roughness, oxide content, and galvanic coupling effect. These correlations were elucidated in light of various results obtained from antibacterial and bacterial attachment tests, and materials characterizations of the coatings. It is possible to tailor copper coating characteristics to render them more effective against targeted bacteria.

## 1. Introduction

The antimicrobial behavior of copper has been known for several millennia. Early studies showed that copper was able to inactivate a wide range of bacteria at different rates [1,2]. Recent studies showed that copper is not only capable of killing bacteria but also inactivating viruses such as Influenza A [3], Coronavirus 229E [4] and SARS-CoV-2 (COVID-19) [5,6].

Vincent et al. [7] proposed that the copper ions on the copper surface break down the bacterial cell membrane and damage their DNA. Factors attributed to the bactericidal properties of copper are summarized in Table 1. High purity copper exhibited greater bacteria inactivation than alloys [7] as undiluted copper releases more copper ions in Cu(0) form to destroy bacterial membranes. Pure copper in a plate or thin sheet form evidently kills both Gram-positive and Gram-negative bacteria [8,9,10,11]. Moreover, it has been reported that copper oxide also kills both types of bacteria when present in powdered form [12,13].

Copper coatings manufactured by various technologies have shown antibacterial efficacy. Thin films of copper fabricated by chemical vapor deposition manifested bactericidal capability yielded by Cu(0) and Cu(I) species [11]. These copper species are also attributed to the efficacy of copper coatings manufactured by the wire arc spray [14]. In addition to copper species, Sharifahmadian et al. [15] demonstrated the influence of surface roughness on the bactericidal efficacy of wire arc-sprayed copper coatings. One recent study by Victor et al. [16] re-iterated the relationship between copper coating hardness—produced by strain hardening during cold spray deposition, with bacterial killing properties.

Studies conducted by different research groups on the bactericidal efficacy of copper were performed independently using different types of bacteria and different forms of copper manufactured by several technologies (Table 1). However, from the current literature review, conclusive knowledge regarding factors influencing the antibacterial efficacy due to the differences in copper coating conditions is lacking.

Copper is a metal element with a density of 8.96 g/cm^3^ [20], higher than other common metals, such as aluminum (Al) with a density of 2.7 g/cm^3^ [20] and iron (Fe) with a density of 7.87 g/cm^3^ [20]. With the same volume, a component made from copper is more than twice as heavy as Al and one and a half times heavier than iron. Due to this limitation, it is preferred to generate copper coatings on a component surface to exploit its bactericidal power without significantly altering the overall component weight. Although silver is another metal with very good antibacterial efficacy against *S. aureus* and *E. coli*, copper is present abundantly and is a lot cheaper coating material than silver [21,22].

A recent development in additive manufacturing technology promoted a cold spray (CS) process as a means for surface functionalization by generating coatings for the energy and environmental industry [23]. In addition, thin layers (90 µm) of Ni manufactured by this technology also showed the capability of enhancing the bonding strength of stir butt-welded dissimilar metals such as Al and Cu [24]. Cold spray technology uses compressed and heated carrier gas—helium, nitrogen or air, to increase micro-to-nano scaled solid particles velocity to a level higher than the critical one before particle-substrate and particle-particle impact occurs. Particle velocity determines the particle-substrate adhesion and particle-particle cohesion strength after impact. Experiments and numerical simulations using finite element analysis [25,26,27] and molecular dynamics method [28] confirmed the influence of critical velocity for different types and sizes of metal particles on the deposition efficiency. Furthermore, since CS technology operates at a temperature below the melting point of the particle, constraints associated with high-temperature processing, such as phase transformation and oxidation, can be kept to a minimum. Furthermore, CS technology is capable of manufacturing complex geometry coatings in a relatively short time, owing to its high build rate [6,29].

This study aims at understanding the effect of copper species, surface roughness and hardness on the bactericidal efficacy posed by cold-sprayed copper coatings through a holistic approach. Bacterial membrane integrity and damage are probed through scanning electron microscopy. The understandings gained from this study will aid in developing solutions to fight bacterial contamination via cold-sprayed coatings.

## 2. Materials and Methods

### 2.1. Materials

Copper powder with chemical composition presented in Table 2 was used to manufacture coatings on stainless steel (SS304) and aluminum (Al5005) substrate plates. The powders consisted of spherical, elongated and irregular morphology particles, with sizes between 5 and 60 µm, as shown in Figure 1.

### 2.2. Cold Spray Coatings

Cold spray deposition was performed using a novel system where a stationary nozzle deposits metal powder on a moving substrate. Metal powder in the feeder is fed through a tube using a low-pressure line while simultaneously heated high-pressure air flows into the nozzle. Subsequently, metal powder and heated compressed air are mixed and deposited through a nozzle onto a substrate attached to the end effector on the robot arm (Figure 2). In this system, the motion and speed of the substrate attached to an effector of a six-axis robotic arm are controlled by a toolpath algorithm generated by a proprietary simulation software, TwinSpee3D^®^ (Spee3D, Dandenong, Victoria, Australia). Hutasoit et al. [30,31] presented a more detailed description of this deposition system in their earlier work.

In this study, copper powder was deposited onto SS304 and Al5005 substrate at 45° to the nozzle axis in conjunction with 16 mm stand-of-distance, 30 bar air pressure and 500 °C air temperature, to produce 60 × 60 × 0.7 mm coatings on 80 × 80 × 2 mm SS304 and 80 × 80 × 3 mm Al5005 substrates.

The copper coatings on stainless steel were tested under two conditions: the as-deposited (AD) state and the other annealed at 400 °C for 10 min to induce an oxide layer on the surface of the copper deposits (HT state). Heating copper deposit at 400 °C—a temperature at the lower end of copper oxidation [32,33], was expected to induce Cu_2_O and CuO layers on copper deposit. The copper-coated aluminum samples were tested in the AD condition only. A wrought sample of copper (bulk Cu) was also tested for comparison with coated samples. The various sample conditions studied are presented in Table 3. Following the manufacturing of copper deposits on SS304 and Al5005, small samples with a 10 × 10 mm dimension were sectioned from each copper-coated and bulk Cu plate prior to conducting the antibacterial and other tests.

### 2.3. Material Characterization

Surface roughness was measured using a profilometer at three different locations on as-built and heat-treated copper coated and bulk copper samples. One sample from each condition was mounted in bakelite followed by wet grinding with water-proof SiC papers to 2000 grit and polished with 9 μm, 3 μm, and 1 μm diamond suspension solutions. Hardness testing was carried out on polished samples using Buehler Micromet 3 micro-Vickers hardness tester with a diamond indenter using 5 gf load (HV_0.05_) with a dwelling time of 10 s at ten random locations on the coating surface. For microstructure analysis, polished specimens were etched with a reagent containing 2 g ferric chloride, 40 mL distilled water, and 10 mL concentrated hydrochloric acid to reveal the particle and grain boundaries in each sample. The microstructure was studied using an optical microscope and ZEISS SUPRA 40VP scanning electron microscope (SEM) equipped with secondary and backscatter electron detectors.

For phase identification and quantification, X-ray scans were performed on the deposit surface from each condition. A Bruker D8 Advance X-ray Diffraction (XRD) machine operating at 40 kV and 30 mA equipped with a graphite monochromator, a Ni filtered Cu Kα (*λ* = 1.5406 nm) source, and a scintillation counter was used to obtain the XRD spectra for each sample. Quantitative phase analysis was performed using MAUD software version 2.97 (http://maud.radiographema.eu/).

### 2.4. Antibacterial Tests

The antimicrobial efficacy of the above-mentioned Cu samples (10 × 10 mm dimension) was assessed against *Staphylococcus aureus* (ATCC 25923) and *Escherichia coli* (ATCC 25922). Before antimicrobial testing, all the samples were sterilized by immersing in 70% ethanol (Sigma-Aldrich, Australia) for 5 min, then air-dried under aseptic conditions. The test organisms were grown on Brain Heart Infusion (BHI) agar (Edwards, Australia) overnight at 37 °C. One bacterial colony of either bacteria was inoculated into 10 mL BHI broth (Edwards, Australia) and incubated overnight at 37 °C. The turbidity of the bacterial cell suspension was measured at 600 nm using a Helios Epsilon spectrophotometer and was adjusted to an optical density that corresponded to approximately 10^8^ colony forming units (cfu) per mL. 1 × 1 cm microscopic glass slides (Livingstone, Australia) were used as control surfaces in this study. 5 µL of the adjusted cell suspension was spotted on each of the different Cu samples and on control glass slides in triplicate. Spotted bacterial suspensions were air-dried for 5 min and incubated aseptically at ambient temperature for allocated exposure times (0, 15, 30, 45, and 60 min). The samples were then transferred into 1 mL sterile phosphate-buffered saline (PBS) and vortexed for 30 s to recover the adhered bacterial cells. All samples were carefully examined to ensure that the bacteria had been successfully detached from the surface, as per ASTM E2197-17e1 [34]. Serial dilutions of the resulting suspensions were prepared in PBS, and 100 µL from each dilution was spread on BHA plates (in triplicate). Plates were incubated overnight at 37 °C, and the resulting bacterial colonies were used to determine the viable counts (expressed as log_10_ cfu/mL). Finally, log reductions were calculated by subtracting the viable counts of bacteria exposed to Cu surfaces from those of bacteria recovered from glass slides. This is an EPA validated procedure [35] and a similar antibacterial testing technique and incubation time has been used by other researchers [1,17,36,37].

### 2.5. Bacteria Attachment Tests

Bacterial attachment tests were conducted on Cu samples, followed by examination using an SEM at different magnifications between 1000x and 20,000x. Cu samples used for the attachment studies were Bulk Cu, Cu/SS-AD and Cu/SS-HT. Glass slides were used as a control in the attachment tests. The primary purpose of the attachment test was to analyze differences in the morphology of bacterial cells attached across different samples. Before the attachment test, the samples were sterilized by immersion in 70% ethanol (Sigma-Aldrich, Sydney, Australia) for 5 min and then air-dried aseptically in a laminar flow cabinet. The bacteria (i.e., *S. aureus* and *E. coli*) were cultured overnight in BHI broth at 37 °C. The turbidity of bacterial cell suspension was adjusted to an optical density corresponding to 10^8^ cfu/mL. A 15 µL sample of adjusted cell suspension was placed on each of the three different Cu samples tested in duplicate and incubated at room temperature for 15 min. The samples were then gently rinsed with sterile PBS and fixed using a 3.0 vol.% solution of glutaraldehyde in PBS for 30 min at room temperature. The coupons were then washed twice with sterile deionized water followed by stepwise dehydration with 25%, 50%, 75%, 90%, and 100% ethanol for 10 min each. The coupons were aseptically dried in a laminar flow cabinet before being finally examined under SEM. The glass slide samples are non-conductive and hence were gold coated before SEM examination. The images of bacterial cells attached to control glass slides and different Cu samples were taken to observe their morphology, and their dimensions were determined.

## 3. Results

### 3.1. Antibacterial Testing

The antibacterial properties of different copper (Cu) samples were assessed against *S. aureus* and *E. coli*. The viable counts of recovered cells from glass slides at different time points were approximately 10^5^ cfu/mL. Therefore, the maximum log reduction calculated for each sample type at different time points was (5), which indicated no recovery of viable bacteria.

The results shown in Figure 3 and Table 4 demonstrate that all the Cu samples have antimicrobial activity against *S. aureus* and *E. coli*. For both *S. aureus* and *E. coli*, Cu/SS-HT samples showed reduced killing efficiency compared to Cu/SS-AD and Cu/Al-AD and the bacterial cells were not eliminated even after 60 min of incubation. On the other hand, Cu/SS-AD showed complete killing at 30 min for *S. aureus* and 15 min for *E. coli*, while Cu/Al-AD eliminated all viable *S. aureus* and *E. coli* at 0 min and 30 min, respectively.

### 3.2. Bacterial Attachment Testing

The SEM images of bacterial cells attached to the control glass samples are shown in Figure 4 which clearly demonstrated the spherical cocci-shaped *S. aureus* cells attached in groups resembling grape-like clusters, and rod-shaped *E. coli* cells attached individually.

Figure 5 shows the SEM images of the *S. aureus* and *E. coli* bacterial cells attached to the different Cu samples tested, i.e., bulk Cu, Cu/SS-AD and Cu/SS-HT. The inset within each separate image shows the high magnification image of bacterial cells attached to the surface and is highlighted with a red square box. The other locations of bacterial cells attached to the surface of the samples are indicated by yellow arrows, whereas the blue arrows are used to indicate the cell debris produced by the bacterial cells. Although the morphology of the *S. aureus* and *E. coli* bacterial cells attached to these Cu surfaces was consistent with that observed on the control glass samples, Cu surfaces exposed to *S. aureus* showed a higher amount of cell debris than Cu surfaces exposed to *E. coli*. The average dimensions of bacterial cells (n = 15) attached to the glass slides and different Cu samples were determined and shown in Table 5. The results showed no significant difference in the bacterial cell dimensions attached to the surface of the glass, bulk Cu, Cu/SS-AD and Cu/SS-HT.

Bacterial cells exposed to different Cu samples showed several visible changes (shown in Figure 6) as compared to cells on the glass samples. Most of the grape-bunch group of *S. aureus* cells on bulk Cu, Cu/SS-AD and Cu/SS-HT were ruptured and deformed. An increased amount of cell debris was also evident on Cu-coated samples, thus indicating these Cu samples might have major effects on the cell wall or cytoplasmic membrane of Gram-positive bacteria. High magnification SEM images of *E. coli* (Figure 6) also showed atypical variations in cell morphology with compromised integrity of cell membrane, causing shrinkage and deformation of cells.

### 3.3. Copper Coating and Bulk Copper Characterization

XRD spectra for bulk copper and copper coatings are shown in Figure 7. Bulk and as-deposited (AD) copper, regardless of the type of substrate coated on, show predominantly pure copper. However, a significant difference was evident in heat-treated copper deposited on SS304, where the presence of Cu_2_O and CuO phases were observed due to heat treatment at 400 °C for 10 min. The Cu_2_O and CuO content on the surface of the heat-treated copper deposit was found via quantitative phase analysis to be 25.71 and 11.54%, respectively.

Coating cross-sections (perpendicular to the deposition direction) and their properties are presented in Figure 8 and Figure 9, respectively. Cold sprayed coatings showed significant porosity irrespective of the coating state. The presence of porosities near the surface and within the cold sprayed coatings is a common occurrence and is attributed to the inter-particle bridging phenomenon [38]. The Cu/Al-AD samples showed the highest coating thickness and porosity content, followed by Cu/SS-AD and then Cu/SS-HT (Figure 9c,d). The Cu/SS-HT coating showed reduced thickness and porosity, indicating coating densification during heat treatment [39].

In Cu/SS-AD samples, coating delamination caused by inadequate particle-substrate adhesive strength was observed (Figure 8a). This phenomenon typically occurs as a result of depositing soft particles on a hard surface, such as copper (typical hardness of 130 HV) on stainless steel SS304 (typical hardness of 200 HV), as was the case in this study. Soft particles are incapable of penetrating hard substrates to establish adequate particle-substrate interlocking, and therefore, are prone to delamination. Moreover, the same amount of Cu powder was deposited onto both SS304 and Al5005 substrates; however, the coating thickness of the Cu/SS samples was lesser than Cu/Al samples (Figure 9c), indicating reduced deposition efficiency when coating Cu on SS substrates.

When the Cu/SS samples were heat-treated, the effect of delamination decreased, nonetheless, the coating/substrate interface was clearly distinct (Figure 8b). This indicated that the heat treatment temperature of 400 °C was insufficient to create a metallic bond between Cu coating and the SS substrate.

In the as-deposited state, Cu/Al-AD generated a thicker coating since the Al5005 substrate is relatively softer (typical hardness of 100 HV) than the Cu powder particles. Therefore, it is easier to deposit Cu coating on softer Al5005 substrates compared to the harder SS304 substrates. This is evident from the extremely well bonded Cu/Al coating interface as seen in Figure 8c, which becomes an anchor for the subsequent layers to build upon, thereby generating thicker coating and increased deposition efficiency.

Cold spray coatings generally have a rough surface compared to wrought samples, which was evident in this study as well (Figure 9a). Bulk Cu exhibited lowest surface roughness. The surface roughness of the as-deposited samples (Cu/SS-AD and Cu/Al-AD) was almost similar, however, higher than bulk Cu samples, whereas Cu/SS-HT samples showed the higher roughness.

The microstructures of the samples are shown in Figure 10. Bulk Cu exhibited the largest grain size compared to the cold sprayed samples (Figure 10a). This can be attributed to the thermomechanical treatment experienced by the wrought Cu billets during processing, which allows recrystallization and considerable grain growth. On the contrary, the as-deposited Cu coatings (Cu/SS-AD and Cu/Al-AD) exhibited extremely deformed powder particles (Figure 10b,d) which was the result of intense dynamic deposition pressures occurring during the cold spray process. Owing to the deformed and strain-hardened powder particles formed during the cold spray deposition, the as-deposited samples exhibited the highest hardness, as shown in Figure 9b. However, when Cu/SS-AD samples were subjected to annealing at a temperature significantly higher than the recrystallization temperature of Cu (~200 °C), the microstructure transformed from deformed powder particles to heavily recrystallized grains (Figure 10c), resulting in decreased hardness compared to as-deposited samples (Figure 9b).

## 4. Discussion

### 4.1. Effect of Cu Ionic Species on Antibacterial Efficacy

Figure 11 shows the effect of copper oxide content on the inactivation time of both *S. aureus* and *E. coli* bacterial cells. The high concentration of Cu(0) (Cu(0) ions are exhibited by metallic copper, in the absence of copper oxides) ionic species in bulk and as-deposited conditions (Figure 7) resulted in considerably high antibacterial efficacy wherein both types of bacterial cells were inactivated within 30 min of exposure to these surfaces, as reported in Table 4. Conversely, the annealing of Cu/SS-AD resulted in the formation of Cu_2_O (Cu(I) ionic species) and CuO (Cu(II) ionic species) oxides on the surface of Cu/SS-HT, with a total oxide content of about 37% (Figure 7). Both bacterial cells when exposed to Cu/SS-HT surfaces did not inactivate completely even after 60 min exposure time (Figure 11a). This strongly indicates that the lower concentrations of Cu(0) in the Cu/SS-HT coatings, along with the presence of Cu(I) and Cu(II) ionic species, results in reduced bacteria-killing ability.

It is well understood that copper has very good antibacterial properties against a variety of bacteria wherein the Cu ions react with lipid causing peroxides of membrane phospholipids resulting in the loss of membrane integrity and bacterial cell death [40,41]. Bacterial cells exposed to Cu surfaces in buffer were killed in hours, whereas the microbes were inactivated within minutes when exposed to dry Cu surfaces [36,42,43]. This is due to the fact that the efficacy of Cu(I) ions is slightly lower than Cu(0) ions, as has been reported by several studies [11,12,14,44].

In this study, it was found that the presence of Cu(II) ions further exacerbated the bacterial killing properties of the copper coatings. Similar results were observed by Mazurkow et al. [45]. To explain this phenomenon, schematic representations are depicted in Figure 11b,c, which shows that in the case of as-deposited Cu coatings, the Cu(0) ions readily bind with both types of bacterial cells. However, in the presence of copper oxides which release Cu(I) and/or Cu(II) ions, there are competing effects between the various Cu ions. Cu(I) and Cu(II) ions have a lesser influence on damaging the bacterial cell membranes [41,45], whereas a lower concentration of Cu(0) leads to reduced interaction of Cu ions with the bacterial cells. Therefore, it is evident that the presence of copper oxides (either in the form of Cu_2_O or CuO) is deleterious to the antibacterial property of Cu coatings.

### 4.2. Effect of Cu Coating Properties on Antibacterial Efficacy

The effect of different Cu coating properties, including porosity, thickness, hardness, and surface roughness on the inactivation capability of the bacterial cells (both *S. aureus* and *E. coli*) was analyzed and presented in Figure 12. The Cu/SS-HT surface did not completely inactivate both bacterial cell types after an exposure of 60 min and the reasons for this occurrence have been explained in Section 4.1. Moreover, the Cu/Al-AD surface immediately inactivated *S. aureus* cells (i.e., at 0-time interval of exposure). The probable explanation for this result is discussed in Section 4.3. Therefore, these data points were considered ‘outliers’ in Figure 12. Although there are limited data, generic trends between these parameters could be examined.

From Figure 12, the following trends can be noted distinctly:The inactivation time for both bacterial cells increased with an increase in coating porosity, coating hardness, and surface roughness of the coatings.The inactivation time for both bacterial cells decreased with an increase in coating thickness.

As expected, the increase in coating porosity resulted in a decrease in the release of Cu ions, thereby resulting in reduced antibacterial efficacy. Contrarily, the increase in coating thickness induced a higher release of Cu ions, thereby resulting in increased antibacterial efficacy.

With regards to the hardness of the coating and its influence on antibacterial efficacy, several research studies [16,18,46,47] have demonstrated that copper particles with a higher hardness level corresponded to a high dislocation density which increased the diffusion of copper ion and therefore, increased the microbial killing efficiency. However, in this study it was found that, although the as-deposited coatings had higher hardness values (130–140 HV) due to severe powder particle deformations during the cold spray process, they exhibited lower bacterial inactivation capability, contrary to the published observations. Since there are other factors or coating properties that also affect the bacterial efficacy, it is highly likely that the effect of hardness is considerably less than the others.

An increase in inactivation time for rougher surfaces is expected, due to the fact that finer surface induces more extensive bacteria-surface contact for copper ionization [47] that reacts and damages bacterial cell membrane, leading to complete bacteria inactivation [7].

### 4.3. Effect of Galvanic Coupling Due to Different Coating/Substrate Configurations

Cu/Al-AD exhibited the highest anti-*S. aureus* efficacy compared to those in other conditions; wherein complete bacteria-killing took place at the 0-min time point, indicating immediate inactivation upon *S. aureus* adhering to the copper coating surface (Figure 3). However, when *E. coli* was exposed to Cu/Al-AD coating surface, it took about 30 min to inactivate these bacterial cells, in which case the Cu/SS-AD coating surface performed better to curb the microbial activity. This is contrary to the results previously observed by Jing et al. [48], where the inactivation time for *E. coli* was shorter than *S. aureus* when exposed to porous Cu materials. Therefore, these discrepancies can be directly related to the differences in the intrinsic characteristics of the individual type of bacterial cells and their interaction with various Cu materials/coatings.

It is well known that *E. coli* is a Gram-negative bacterium with a thin peptidoglycan cell wall and an additional outer membrane, while *S. aureus* is a Gram-positive bacterium with a thicker cell wall made of many layers of peptidoglycan [49]. As observed in Figure 4, *E. coli* bacterial cells have a rod-shaped morphology whereas *S. aureus* bacterial cells have a cocci-shaped morphology and are often present as grape-like clusters. *E. coli* is commonly associated with urinary tract infections and bacteremia, whereas *S. aureus* causes several infections, including those of skin and bone, as well pneumonia [50].

In this study copper was coated on two different materials, one being stainless steel and the other aluminum. This induces galvanic coupling effect which occurs between two dissimilar metals in contact with each other under electrolytic conditions, wherein one metal corrodes/reacts preferentially than the other metal. The steady-state potential which is defined as negative to a saturated calomel half-cell of metal and alloys listed in a galvanic series based on potential measurements in flowing seawater at 25 °C, was referred [51]. The steady-state potential of stainless steel (300 series), copper, and aluminum are 0.08, 0.36, and 0.79 V, respectively [51]. Therefore, the galvanic potential difference between copper and stainless steel (Cu/SS) is 0.28 V, and that of copper and aluminum (Cu/Al) is −0.43 V. Cu/SS having a positive galvanic potential reacts better with Gram-negative *E. coli* bacterial cells, inactivating them in 15 min. However, when the *E. coli* cells were exposed to copper coating surface on Cu/Al samples (which has a negative galvanic potential), they were inactivated in 30 min, longer than that of the Cu/SS surface. Likewise, when Gram-positive *S. aureus* cells were exposed to negative galvanic potential Cu/Al samples, they were inactivated relatively very fast (0-min time point), and when exposed to positive galvanic potential Cu/SS sample surface, they took 30 min to be inactivated completely.

The experimental results from this study suggest that there is a strong correlation between the type of bacteria and the galvanic potential of the copper/substrate bimetallic surfaces. This has been schematically shown in Figure 13. In the case of bulk copper which is in positive steady-state potential, there are more Cu ions released. In this environment, these Cu ions react more aggressively with Gram-negative *E. coli* bacterial cells rather than Gram-positive *S. aureus* bacterial cells. Therefore, the inactivation time for *E. coli* on bulk copper was 0-min time point and that of *S. aureus* was 15 min. Many other studies also confirmed better antibacterial response from different copper coatings on Gram-negative bacterial cells [52].

In the case of copper coatings, there are significantly less Cu ions released by the coating surface when it forms a positive galvanic couple with the substrate material, as opposed to the higher amount of Cu ions with a negative galvanic couple between the copper coating and the substrate [45]. When there are less Cu ions present, it results in more time for bacterial inactivation, irrespective of the type of bacteria. Therefore, Cu/SS-AD exhibited an inactivation time of 15 min and 30 min for *E. coli* and *S. aureus*, respectively.

For a Cu/substrate bimetallic configuration with negative galvanic potential, there is a higher amount of Cu ions. However, in this environment, the Cu ions react aggressively with Gram-positive *S. aureus* bacterial cells rather than Gram-negative *E. coli* bacterial cells. Therefore, the inactivation time for *S. aureus* on Cu/Al-AD sample surface was 0-min time point and that of *E. coli* was 30 min.

Gottenbos et al. [53] reported that both types of bacterial cells adhered rapidly to the positively-charged contact surfaces, but once attached the Gram-negative bacteria did not proliferate. In contrast, both types of bacterial cells adhered very slowly to the negatively-charged surfaces; however, once bound, exhibited rapid growth in the number of cells. The authors concluded that the positively-charged contact surfaces exerted an antimicrobial effect only on Gram-negative bacterial cells. This is in line with the observations made in this study. A positive galvanic potential of the copper coated surface leads to good antibacterial efficacy against Gram-negative *E. coli* bacteria, and negative galvanic potential of the copper coated surface results in rapid inactivation of Gram-positive *S. aureus* bacteria.

Based on this information, it is possible to design high-contact components with suitable coatings that respond in an expected manner to several generic microbial cells.

### 4.4. Bacteria Killing Mechanisms

The exact bactericidal mechanism on the various copper surfaces could not be determined through the results obtained in this study. However, it was evident that the presence of Cu ions from various surfaces significantly affected the appearance of both types of bacterial cells (*E. coli* and *S. aureus*), resulting in abnormal shapes, stained cells, disrupted cell membranes, and release of cell debris, as shown in Figure 6. One of the possible mechanisms suggested for killing bacterial cells in contact with the metallic copper surface was that the dissolved copper ions cause cell damage and/or rupture of the cell membrane [1,7]. This explains the effect shown on the bacterial cells in Figure 6.

## 5. Future Works

In this study, the antibacterial responses of copper coatings in both as-deposited and heat-treated states were investigated. The results show that these cold-sprayed copper coatings have good antibacterial efficacy in the as-deposited state. However, in most applications, it is a norm that the coatings are subjected to post heat treatment for generating enhanced mechanical properties. Therefore, further studies need to be carried out to improve the antibacterial performance of heat-treated copper coatings.

From this study, it was found that there is a very complex relationship between various coating properties such as porosity, hardness, surface roughness, and microstructural phases on the antibacterial efficacy. However, further exploration is needed to comprehensively determine the impact of each of these coating properties on the bacterial killing property of the copper coatings.

In future studies, it would be interesting to see the effect of increasing the attachment time (i.e., t > 15 min) on bacterial cell morphology and undertake live/dead cell staining, which will provide further information on the degree of reduction of viable bacterial cells attached to the surface of different copper samples. The interaction of the bacterial cells with various copper coatings should be recorded in-situ real-time. Moreover, only one type of bacterial assay was used in this study. Additional testing with different types of assays is required for a comprehensive understanding of the antibacterial efficacy of cold-sprayed copper coatings.

Lastly, it is well known that the properties of copper coatings degrade over a period of time due to exposure to various environmental factors resulting in oxidation and/or wear. Further investigations are required to determine whether the antibacterial performance of copper coatings deteriorates over time and possibly explore strategies to develop long-lasting copper coatings.

## 6. Conclusions

Copper is well known for its antimicrobial property. It can kill different types of bacteria and viruses upon contact. In this study, the antibacterial responses of copper coatings produced using a cold-spray process, an advanced rapid manufacturing and coating process, was studied, and the results were compared with bulk copper counterparts. The copper coatings were cold-sprayed onto stainless steel (SS304) and aluminum (Al5005) substrate plates. *E. coli* and *S. aureus* bacterial cells were exposed to these samples and their inactivation times were recorded. The results were analyzed and studied in correspondence with other coating properties such as thickness, porosity, hardness, roughness, and heat treatment. The main conclusions that can be drawn from this study are:

The copper coatings in the as-deposited state inactivated both types of bacterial cells under 30 min exposure. However, when the samples were heat-treated, both *E. coli* and *S. aureus* were not completely killed even after 60 min exposure to the copper-coated surfaces.

It was found that there was a presence of copper oxides (Cu_2_O and CuO) on the surface of heat-treated samples. This resulted in reduced antibacterial efficacy due to the presence of Cu(I) and Cu(II) ions and lower concentrations of Cu(0)—metallic copper ions.There was an insignificant difference in the bacterial cell morphology (shape and dimensions) of both *E. coli* and *S. aureus* exposed to different types of copper samples. Moreover, both bacterial cells showed signs of damage resulting in abnormal shapes, stained cells, disrupted cell membranes, and release of cell debris when seen under an SEM.The inactivation time for both *E. coli* and *S. aureus* appeared to increase with an increase in coating porosity, hardness, and surface roughness, but decrease with an increase in thickness of copper material.Cu ions from bulk copper surface react more aggressively towards the Gram-negative *E. coli* bacterial cells than Gram-positive *S. aureus* cells, inactivating *E. coli* faster than *S. aureus*. However, in the case of bimetallic galvanic coupling with a negative potential, such as Cu/Al coating/substrate configuration, it is possible that the Cu ions react more aggressively towards Gram-positive *S. aureus* resulting in shorter inactivation time compared to *E. coli*.Copper coating/substrate configuration with a positive galvanic coupling potential (i.e., Cu/SS in this study) could yield less Cu ions compared to bulk copper samples which resulted in higher inactivation time for both types of bacterial cells.Different coating/substrate galvanic coupling configurations respond differently to different types of bacteria. Therefore, it is possible to tailor coating properties based on various factors such as galvanic coupling, post heat treatment, coating porosity, thickness, etc. to render them effective towards targeted bacteria.

## Figures and Tables

**Figure 1 materials-14-06744-f001:**
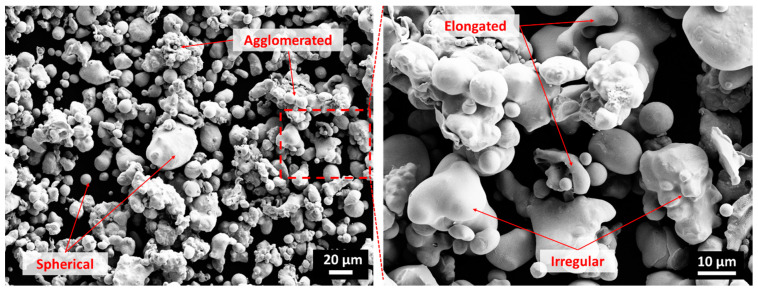
Scanning electron microscope (SEM) images of copper powder consisting of a mixture of spherical, agglomerated, elongated, and irregular-shaped particles.

**Figure 2 materials-14-06744-f002:**
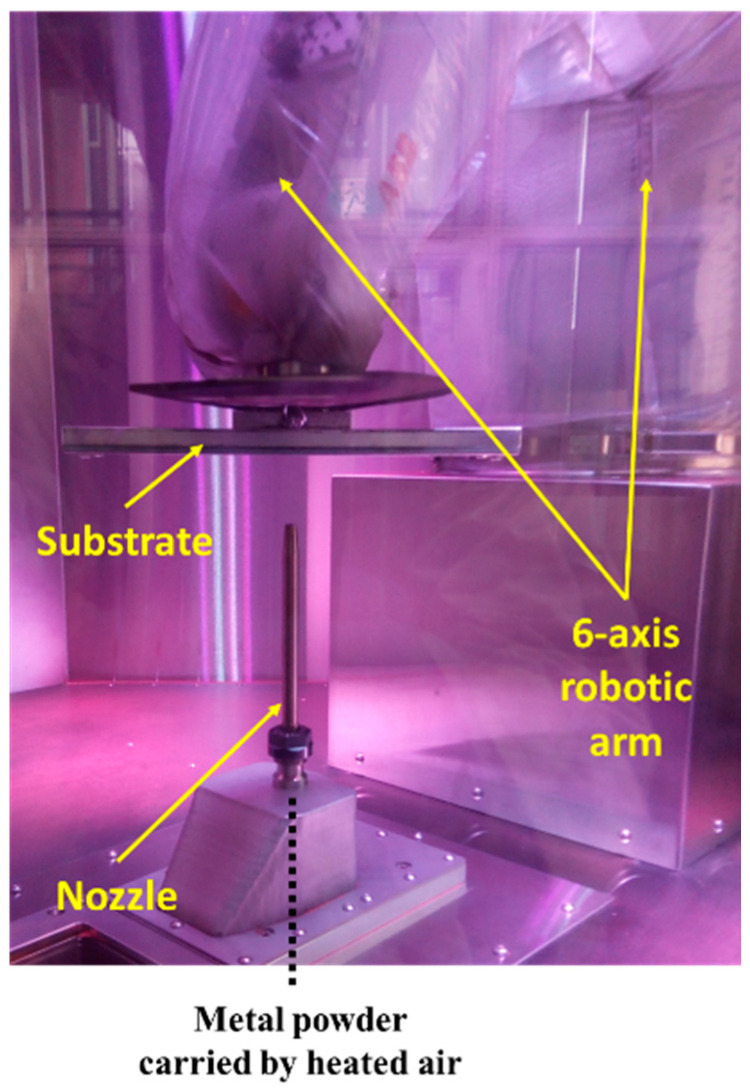
Cold spray coating machine used in this study consisting of a stationary nozzle through which copper powder was sprayed onto stainless steel or aluminum substrates, attached to the end effector of a 6-axis robotic arm.

**Figure 3 materials-14-06744-f003:**
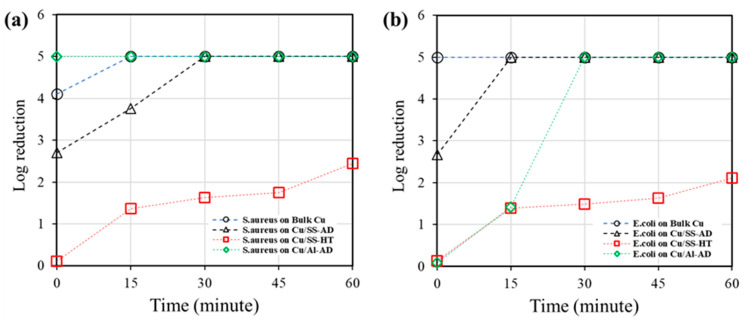
Antibacterial efficacy of bulk Cu and Cu-coated samples against (**a**) *S. aureus* and (**b**) *E. coli.* (data represent average value of log reduction of viable counts of recovered bacterial cells).

**Figure 4 materials-14-06744-f004:**
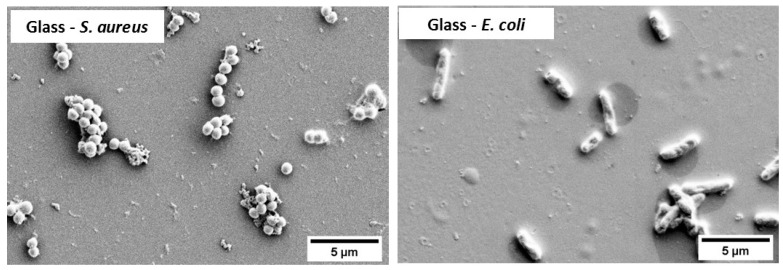
SEM images showing the morphology of *S. aureus* and *E. coli* bacterial cells attached to control glass samples after 15 min exposure.

**Figure 5 materials-14-06744-f005:**
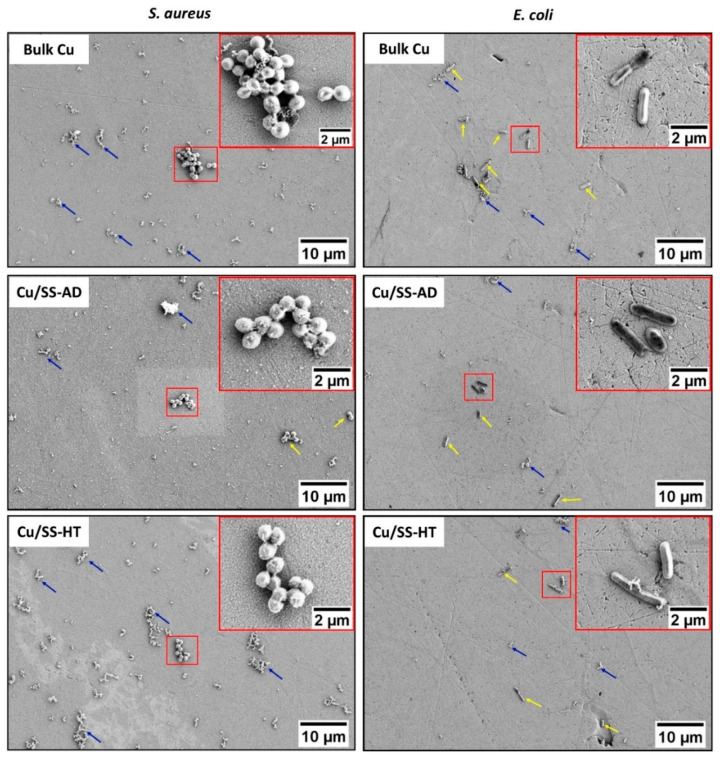
SEM images showing the morphology of *S. aureus* (**left-hand side**) and *E. coli* (**right-hand side**) bacterial cells attached to different Cu samples tested after 15 min exposure. The insets show high magnification images of the bacterial cells. *Note:* Yellow arrows indicate locations where bacterial cells are attached to the surface of the samples, and blue arrows indicate the cell debris produced by the bacterial cells.

**Figure 6 materials-14-06744-f006:**
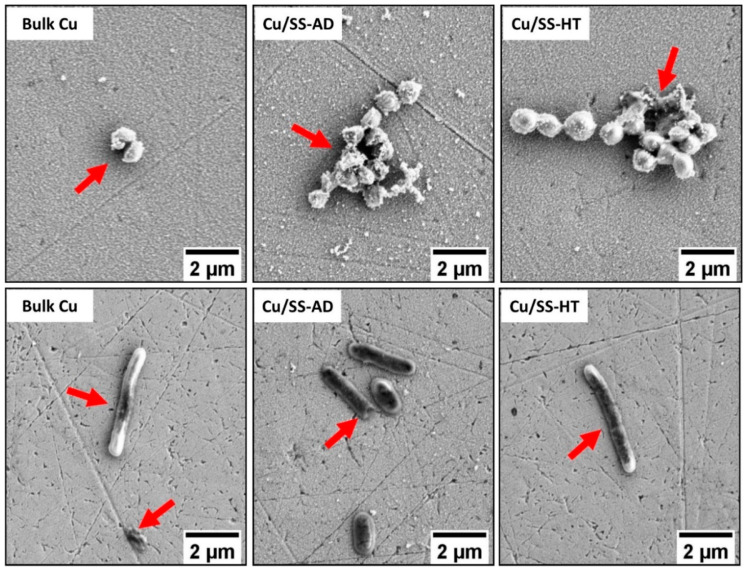
SEM images showing the morphology of bacterial cells *S. aureus* (**top row**) and *E. coli* (**bottom row**) attached to the different Cu samples tested after 15 min exposure. Red arrows are pointing towards abnormal bacterial cell shape/feature and/or disrupted cell walls.

**Figure 7 materials-14-06744-f007:**
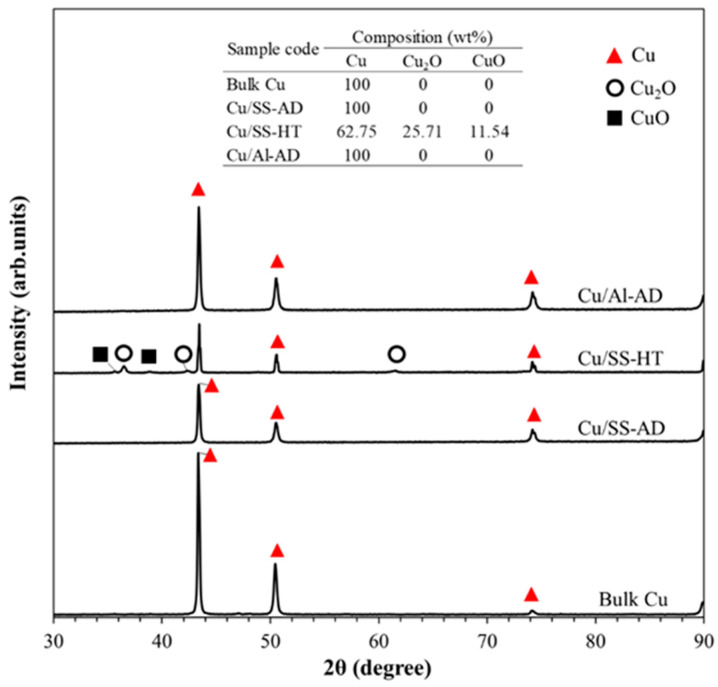
XRD spectra of bulk Cu, Cu/SS-AD, Cu/SS-HT, and Cu/Al-AD samples.

**Figure 8 materials-14-06744-f008:**
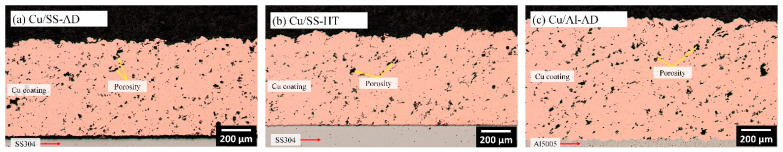
Optical micrographs showing copper coatings on SS304 substrate in (**a**) as-deposited and (**b**) heat-treated states, and on (**c**) Al5005 substrate in as-deposited state. *Note:* Yellow arrows show the porosities in the copper coatings, and red arrows show the substrate.

**Figure 9 materials-14-06744-f009:**
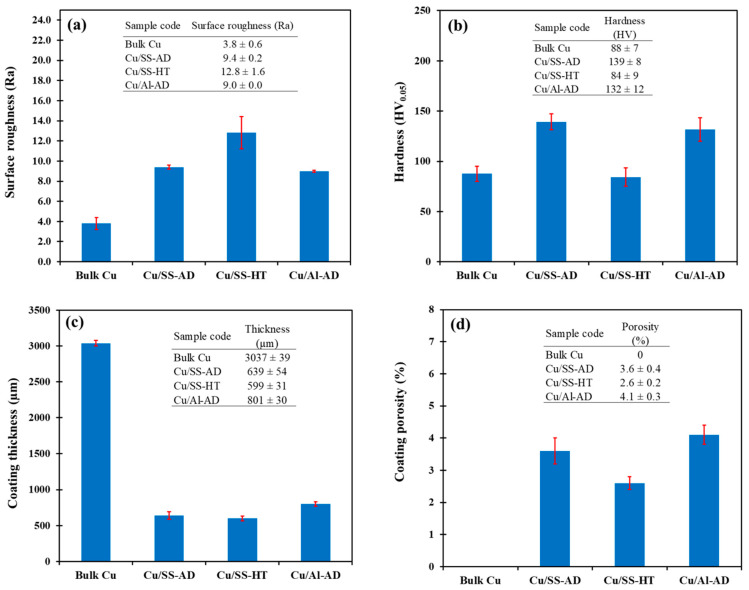
Properties of bulk Cu and Cu-coated samples: (**a**) surface roughness (in μm), (**b**) hardness, (**c**) coating thickness, and (**d**) coating porosity.

**Figure 10 materials-14-06744-f010:**
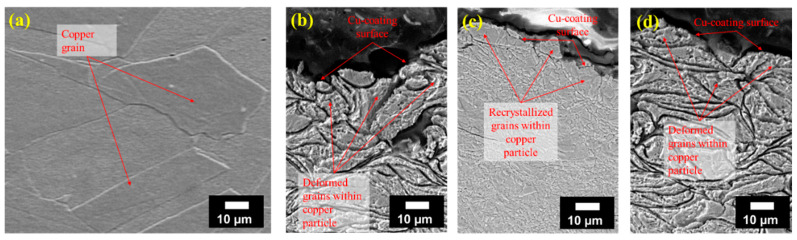
SEM images showing the microstructures of (**a**) bulk Cu, and near-surface coatings of (**b**) Cu/SS-AD, (**c**) Cu/SS-HT, and (**d**) Cu/Al-AD.

**Figure 11 materials-14-06744-f011:**
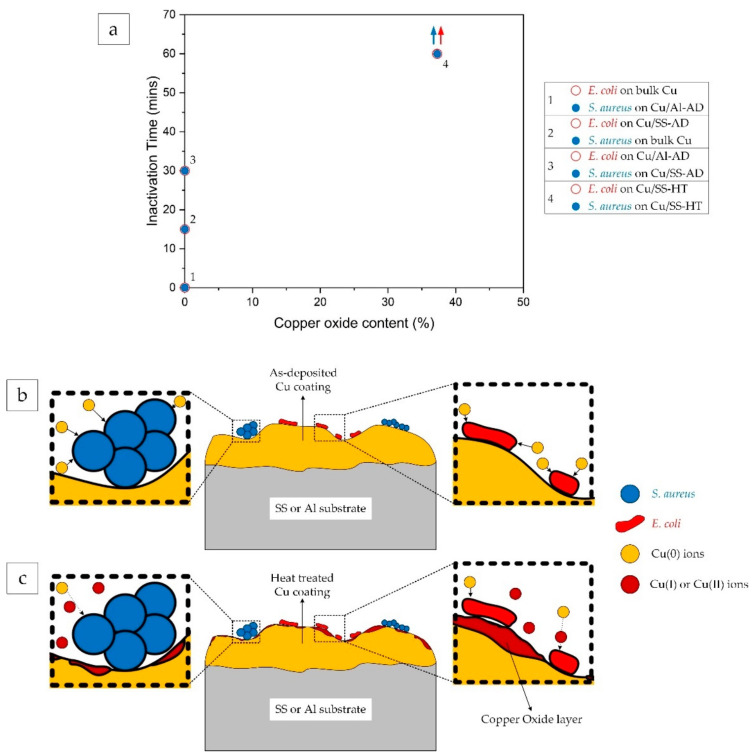
(**a**) Effect of total copper oxide (Cu_2_O + CuO) content on bacteria killing efficacy. The blue and red arrows for *S. aureus* and *E. coli* on Cu/SS-HT indicates that the log reduction of (5) was not achieved after 60 min of bacterial exposure to the surface, (**b**) Schematic of the interaction of Cu(0) ions from as-deposited Cu coating with *S. aureus* (left) and *E. coli* (right), and (**c**) Schematic of the competing interaction of Cu(0), Cu(I), and Cu(II) ions with *S. aureus* (left) and *E. coli* (right).

**Figure 12 materials-14-06744-f012:**
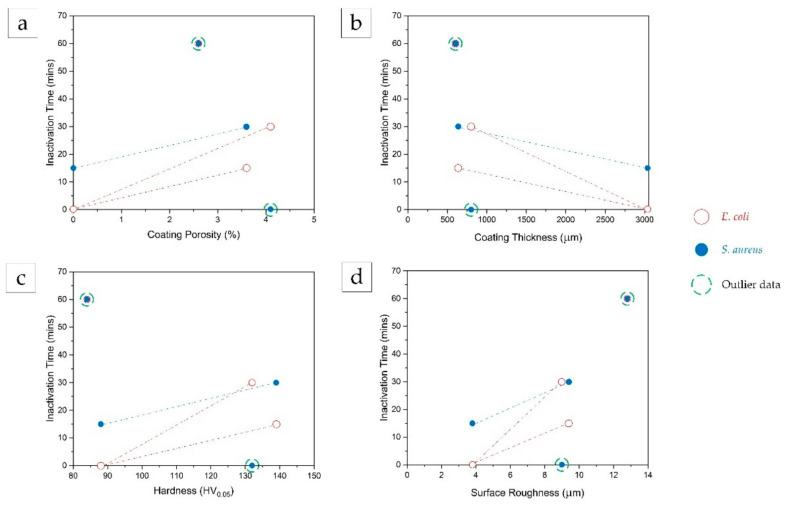
Effect of coating properties, (**a**) porosity, (**b**) thickness, (**c**) hardness, and (**d**) surface roughness, on the inactivation times for *E. coli* and *S. aureus* bacteria.

**Figure 13 materials-14-06744-f013:**
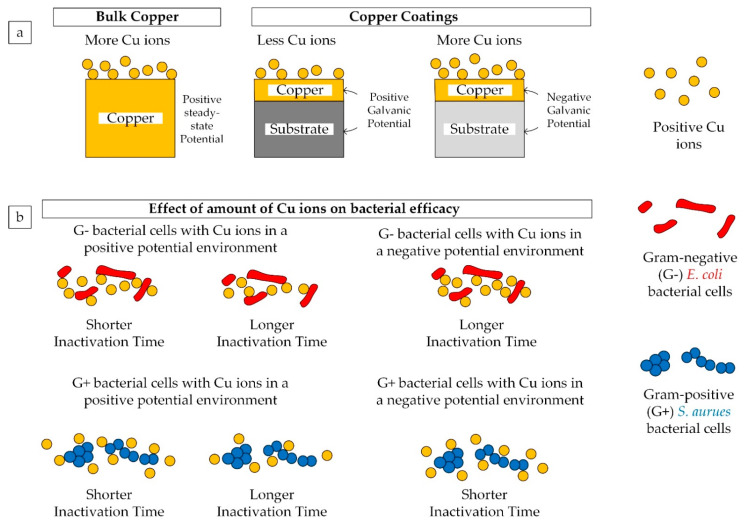
(**a**) Effect of the form of copper and copper/substrate on the amount of Cu ions, and (**b**) Effect of amount of Cu ions on bacterial efficacy.

**Table 1 materials-14-06744-t001:** Antibacterial properties of copper.

Bacteria		Source of Efficacy	Form of Copper	Reference
Species	Cell Wall			
*Escherichia coli* O157	Gram-negative	High copper element content (>70%)	Possibly plate form	[7]
*Acinetobacter baumannii*	Gram-negative			
*Enterobacter* spp.	Gram-negative			
*Klebsiella pneumoniae*	Gram-negative			
*Pseudomonas aeruginosa*	Gram-negative			
*Clostridium difficile*	Gram-positive			
*Enterococcus hirae*	Gram-positive	Cu(0) and growth media	Solid copper, possibly in a plate form	[8]
*Enterococcus hirae*	Gram-positive	Cu(0); Cu(I) found as effective as Cu(0)	Sheet	[9]
*Escherichia coli* (NBRC3972)	Gram-negative	Cu(I)	Powder	[12]
*Staphylococcus aureus*	Gram-positive			
*Escherichia coli* K12	Gram-negative	Cu(0)	Solid copper, possibly in a plate form	[10]
*Escherichia coli* (ATCC 25922)	Gram-negative	Cu(0), Cu(I) found slightly less effective as Cu(0)	Thin film, manufactured via chemical vapor deposition	[11]
*Staphylococcus aureus* (8325-4)	Gram-positive			
*Escherichia coli*	Gram-negative	Cu(I) and Cu(II)	Copper oxides nano-particle	[13]
*Escherichia coli* (W3110)	Gram-negative	Copper ion	Solid copper, possibly in a plate form	[17]
*Bacillus cereus* L8	Gram-positive			
*Deinococcus radiodurans* DSM 20539	Gram-positive			
*Staphylococcus aureus*	Gram-positive	Oxides and surface roughness	Coating, manufactured via wire arc spray	[15]
*Escherichia coli*	Gram-negative			
*Staphylococcus aureus*	Gram-positive	Cu(0) and Cu(I)	Coating, manufactured via wire arc spray	[14]
*Escherichia coli*	Gram-negative			
*P. aeruginosa*	Gram-negative			
Vancomycin-resistant *Enterococcus*	Gram-positive			
Methicillin-resistant *Staphylococcus aureus*	Gram-positive			
*Staphylococcus aureus*	Gram-positive	Strain-hardened particle	Deposit, manufactured via cold spray process	[16,18]
*Escherichia coli*	Gram-negative	Surface roughness	Laser patterning	[19]

**Table 2 materials-14-06744-t002:** Chemical composition of copper powders, stainless steel plate, and aluminum plate, measured by X-ray fluorescence method.

Material	Composition (wt.%)								
	C	Cr	Cu	Fe	Mn	Ni	P	S	Si
Cu powder	-	-	>99.90	-	-	-	-	-	-
SS304	<0.08	18.16	-	71.54	1.07	8.21	-	-	0.48
Al5005	98.89	0.79	0.223	0.041	0.033				

**Table 3 materials-14-06744-t003:** Sample conditions studied.

Sample Code	Condition
Bulk Cu	As-received copper plate
Cu/SS-AD	As-deposited copper coating on SS304 substrate plate
Cu/SS-HT	Copper coating on SS304 substrate annealed at 400 °C for 10 min
Cu/Al-AD	As-deposited copper coating on Al5005 substrate plate

**Table 4 materials-14-06744-t004:** Antibacterial efficacy of various samples against *S. aureus* and *E. coli.* (data represent average value of log reduction ± standard deviation).

Bacteria	Time (min)	Log Reduction			
		Bulk Cu	Cu/SS-AD	Cu/SS-HT	Cu/Al-AD
*S. aureus*	0	4.11 ± 0.17	2.70 ± 0.07	0.11 ± 0.07	(5)
	15	(5) ^1^	3.76 ± 0.10	1.37 ± 0.04	(5)
	30	(5)	(5)	1.63 ± 0.08	(5)
	45	(5)	(5)	1.75 ± 0.02	(5)
	60	(5)	(5)	2.44 ± 0.25	(5)
*E. coli*	0	(5)	2.67 ± 0.07	0.13 ± 0.07	0.09 ± 0.05
	15	(5)	(5)	1.39 ± 0.04	1.42 ± 0.17
	30	(5)	(5)	1.49 ± 0.08	(5)
	45	(5)	(5)	1.63 ± 0.02	(5)
	60	(5)	(5)	2.11 ± 0.25	(5)

^1^ (5) represents the maximum measurable log reduction.

**Table 5 materials-14-06744-t005:** The average dimension of bacterial cells attached to the surface of glass (control) and different Cu samples tested.

Sample Code	*S. aureus*		*E. coli*		
	Shape	Dimensions (μm ± SD)	Shape	Dimensions (μm ± SD)	
		Diameter		Length	Diameter
Glass	Spherical (cocci)	0.9 ± 0.1	Rod	2.2 ± 0.7	0.7 ± 0.1
Bulk Cu		1.0 ± 0.1		2.1 ± 0.4	0.6 ± 0.1
Cu/SS-AD		1.0 ± 0.1		2.4 ± 0.5	0.7 ± 0.1
Cu/SS-HT		0.9 ± 0.2		2.4 ± 0.7	0.6 ± 0.1

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
