# Peer review of "Antibacterial Efficacy of Cold-Sprayed Copper Coatings against Gram-Positive Staphylococcus aureus and Gram-Negative Escherichia coli"

_materials, 2021, doi:10.3390/ma14226744_

Round 1

Reviewer 1 Report

The manuscript described the preparation of Cu- coated materials through an interesting technique and analyses the properties of the materials trying to correlate them with the antibacterial effect.

Unfortunately, the antibacterial experiment is not adequate to address the point. All the conclusions that follow are thus non-significant. A different type of assay should be considered to assess the antimicrobial activity of copper coating. 

The English, the syntax as well as the scientific structure and content of the manuscript are all very poor. 

Scientifically, despite each physiological phenomena occur at incredibly small timescales in bacteria, 15 min-contact is too short to measure bacterial attachment (line 181). In this case Authors might have tested surface stickiness or porosity.

In fact, the antibacterial effects measured by the test herein applied is masked by the propensity of bacteria to adhere to surfaces. Adhesion might prevent their release in PBS and the “log reduction” would appear higher than it actually is. Therefore, a “log reduction” of 5 implies no recovery of viable or detachable bacteria. A more direct test is required. For example, by deposition of the coated-metal foil on agar plates and then assessment of live colonies forming on agar. Anyway, a correlation between these finding and the material porosity is mandatory.

The log reduction graph is complex to interpret and thus poorly informative. Please, find another way to represent the data.

Why residual bacteria are not counted on SEM images?

211: Why Cu/AI-AD samples were not analysed by SEM?

Authors refer to some unknown traces on SEM images as EPS (215/233/467). Are authors aware of what the term “EPS” really means? EPS are NOT cell debris.

The discussion consists of confused over interpretations and imaginative speculations devoid of any scientific base. Also, the writing is even more confused and difficult than in the previous sections. The speculations are not referenced to published work and part of the interpretations are surely unlikely.

318-319/461-463: Have authors considered whether the production of copper oxides decreases the concentration of Cu(0)? Thus, the Copper oxides might not “impede” the bacterial killing properties of Cu(0).

Some (not all) minor comments:

17: solution to IMPROVE/INTEGRATE? antibacterial efficacy.

20: “bacteria” is already plural. The singular is bacterium; their efficacy as?

25: against better than effective

39: unclear. Why “However”? Does this mean that Cu oxide ONLY kills if powdered or in nanoparticles? In that case add ONLY to the sentence.

54: condition is not the most suitable word

62-64: sentence unclear, please rephrase.

62-64: The logical conclusion of this sentence should include a comment on the abundance/value of silver

71: micron or micro?

73: occurs?

75: dynamics?

76: confirmed …the influence/importance of

79-80: owing and equipped… sentence is not coherent

92: comprised of?

118: and after annealing?

194 and in Table 4: “ >” = greater than 5, which is wrong (<5), and anyway inappropriate. Bacteria recovered cannot be less than those that were spotted. Use “at the most/at maximum”

216: Sentence not clear. Please rephrase.

218: higher?

232: pitted? How can this be viewed? One of the images might represent a physiological cell septation…

312: antibacterial efficacy cannot be induced.

314: stress relieving?

317: strains, not cells.

Fig. 11a is a not intelligible.

345: the possible explanation is explained???

Author Response

Please kindly find our responses to your comments in the attached file.

Reviewer 2 Report

The manuscript describes the antibacterial effect of cold-sprayed Copper coatings against aureus and E. coli. It is properly organized and it is well understandable. But some minor revisions are required before publication.

- The size of samples for antibacterial test should be added.

- The authors presented the SEM photograph of samples after antibacterial test, but the photograph of total samples should be also added to see the growth of S. aureus and E. coli. on the samples.

Author Response

(The authors gave the same response as above.)

Reviewer 3 Report

The article is devoted to the study of the applicability of copper coatings for protection against pathogens and various bacteria. The topic of this study is interesting and deserves attention, however, linking to the pandemic and its consequences is not necessary, since the main goal of the work is to study the antibacterial properties of copper coatings, and not to create new ways of protecting against viruses and changing the epidemiological situation. According to the reviewer, the work deserves to be accepted for publication after the authors answer a number of questions that arose during its reading.
1. The choice of the object of research is quite understandable, but the authors should pay attention not only to the positive aspects of copper coatings, but also to the negative ones, which are low resistance to oxidation and degradation of coatings. As is known, copper films oxidize rather quickly in air, which leads to a deterioration in their properties. The authors should provide explanations in this regard.
2. The presented images of coatings show that the surface morphology is prone to pitting corrosion, the authors should explain what this is connected with.
3. The results of X-ray analysis also indicate the presence of oxide phases in the structure of the samples. Authors should give a more detailed explanation of why this is connected.
4. In the conclusion, it is said that the presence of copper oxides contributes to the killing of bacteria, the authors should explain in the article which coatings are better, made of pure copper or with an oxide content.
5. The proposal on the impact of the pandemic should be removed from the abstract, since this work is only indirectly related to the topic of the pandemic.

Author Response

(The authors gave the same response as above.)

Reviewer 4 Report

The manuscript by Novana Hutasoit et al. has investigated Antibacterial efficacy of cold-sprayed Copper coatings against gram-positive S. aureus and gram-negative E. coli.

The study to understand the effect of copper species, surface roughness and hardness on the bactericidal efficacy posed by cold-sprayed copper coatings through a holistic approach. The understandings gained from this study will aid in developing solutions to fight bacterial contamination via cold-sprayed coatings. In general the manuscript contain relevant paragraphs that have been discussed, and the research has been conducted in a proper way.

 After close evaluation of the paper I suggest acceptance of the manuscript.

Author Response

(The authors gave the same response as above.)

Round 2

Reviewer 1 Report

The manuscript has definitely improved in clarity through the revision, however, the conclusion that was already pointed out in the first reviewing process, i.e. (line 331) "This strongly indicates that the presence of  Cu(I) and Cu(II) ionic species impedes the bacterial killing property of Cu(0) ionic species" and (501) "...release of Cu(I) and Cu(II) ionic species which inhibited the high bacterial killing ability of Cu(0) ions" is still far fetched and devoid of any biological support. How can ions inhibit ions? This interpretation is purely imaginative and non-realistic.

In their reply, authors comment that they have no support for alternative explanations, but they neither have support for this one.

The possibility that the effect linked to the lower concentration of Cu(0) is so much more likely than the one proposed and, at least, has a biological basis.

Also, working with biological samples, the requirement is to perform at least three independent and separate replicates of each experiment. Authors declare to have performed two replicas; however no statistical analysis (such as a standard deviation) has been presented.

The lack of repetition of the experiments leads to forced interpretations. As for example in the sentence: (360-362) “Therefore, these data points were considered ‘outliers’ as they did not fit in any trend for  the other results observed in this study. Although there is very little data, but generic trends between these parameters could be examined”, which, besides being grammatically wrong, points to a severe scientific fault: results that do not fit in the model are excluded. So, it is not the model fitting to the data, but the data fitted to a model. This is not science.

Only by comparing average and SD it is possible to judge if the data are reliable.

Possibly, this paradox as well as the unconvincing story of the galvanic potential could be resolved by repeating the experiments, which is a MUST for biological samples. 

Minor comments:

Sentences in lines 348-349, 360-362 need revision

The sentence in lines 409-411 must be rephrased because, as such, is obscure. Furthermore, potentials require a minus sign.

402: remove membrane (which is different from the cell wall). Actually, gram negatives have a thinner cell wall but also an external membrane, lacking in gram positives.

Fig. 13 quality is low. Shorter instead of low- and longer instead of more-inactivation time.

Author Response

Please find the response letter attached.

Reviewer 3 Report

The authors have made corrections in accordance with all the previously mentioned remarks, the article can be accepted for publication.

Author Response

(The authors gave the same response as above.)
